# Volitional EMG Estimation Method during Functional Electrical Stimulation by Dual-Channel Surface EMGs

**DOI:** 10.3390/s21238015

**Published:** 2021-11-30

**Authors:** Joonyoung Jung, Dong-Woo Lee, Yong Ki Son, Bae Sun Kim, Hyung Cheol Shin

**Affiliations:** Electronics and Telecommunications Research Institute, Daejeon 34129, Korea; joonyoung@etri.re.kr (J.J.); hermes@etri.re.kr (D.-W.L.); handcourage@etri.re.kr (Y.K.S.); bskim72@etri.re.kr (B.S.K.)

**Keywords:** electromyography (EMG), EMG-controlled FES, functional electrical stimulation (FES), m-wave, stimulus artifact, volitional EMG (vEMG)

## Abstract

We propose a novel dual-channel electromyography (EMG) spatio-temporal differential (DESTD) method that can estimate volitional electromyography (vEMG) signals during time-varying functional electrical stimulation (FES). The proposed method uses two pairs of EMG signals from the same stimulated muscle to calculate the spatio-temporal difference between the signals. We performed an experimental study with five healthy participants to evaluate the vEMG signal estimation performance of the DESTD method and compare it with that of the conventional comb filter and Gram–Schmidt methods. The normalized root mean square error (NRMSE) values between the semi-simulated raw vEMG signal and vEMG signals which were estimated using the DESTD method and conventional methods, and the two-tailed t-test and analysis of variance were conducted. The results showed that under the stimulation of the gastrocnemius muscle with rapid and dynamically modulated stimulation intensity, the DESTD method had a lower NRMSE compared to the conventional methods (*p* < 0.01) for all stimulation intensities (maximum 5, 10, 15, and 20 mA). We demonstrated that the DESTD method could be applied to wearable EMG-controlled FES systems because it estimated vEMG signals more effectively compared to the conventional methods under dynamic FES conditions and removed unnecessary FES-induced EMG signals.

## 1. Introduction

Functional electrical stimulation (FES) induces the action potential of muscles by stimulating muscle motor nerves with electric current pulses. It has been widely utilized in various areas of rehabilitation, e.g., hand mobility rehabilitation for stroke patients [1], dorsiflexion rehabilitation for foot drop patients [2], and rehabilitation for spinal cord injury patients [3,4,5]. However, FES systems typically require the assistance of physical therapists for controlling the systems or applying predetermined stimulation patterns. To ensure the availability of these systems in real life and enhance their functionality, a patient-active FES system that can be operated in response to the motion intention of a patient has been studied using wearable motion sensors [6,7,8] (e.g., inertial measurement units, accelerometers, encoders, and goniometers), gait phase detection sensors for lower limbs (e.g., foot switch [9]), and muscle activity measurement sensors (e.g., electromyography (EMG) [10], and mechanomyography [11]). Surface EMG sensors are one of the representative muscle activity measurement sensors. They have been mainly utilized to identify the motion intention in patient-active FES systems on the basis of the characteristics of muscle contraction biosignals that can be achieved prior to performing the motion.

There are two types of wearable FES systems integrated with EMG sensors: EMG-triggered [12] and EMG-controlled systems [13]. An EMG-triggered FES system adopts an open-loop FES control algorithm that applies a predetermined stimulus pattern when the amplitude of EMG signals exceeds a threshold value. This method enables the initiation of the stimulus to be intentionally controlled in a simple yet intuitive manner, but it rarely allows the EMG reflecting the motion intention to be measured owing to the difficulty caused by electrical stimulation current of FES. By contrast, an EMG-controlled FES system adopts a closed-loop FES control algorithm in which an EMG value is utilized as an input to modulate a stimulus pattern. This is a completely intentional system because a patient can control not only the initiation and termination of FES, but also the stimulus intensity during stimulation in real-time.

Wearable EMG-controlled FES systems have been extensively studied owing to these advantages. However, EMG signals are inevitably contaminated with stimulus artifacts and the M-wave, which are caused by stimulation current and stimulated muscles when FES and EMG are simultaneously applied to the same muscle. Therefore, to estimate a clean volitional EMG (vEMG) signal, it is crucial to eliminate unnecessary signals from a raw EMG signal. Figure 1 shows a comparison of the EMG data measured with and without FES during periodic muscle contractions. The EMG data are obtained using Delsys Trigno, which is a high-performance device that is considered to be the gold standard for measuring EMG signals [14]. The applied electrical stimulation intensity and frequency are 3 mA and 20Hz, respectively, and the EMG data are sampled at 2000Hz. Regardless of the performance of the EMG system, the EMG data measured with FES clearly show that voluntary muscle contraction information is completely contaminated compared to the EMG data measured without FES, even though the electrical stimulation current is relatively low.

In the early research on estimating vEMG signals during FES, researchers assumed the M-wave as a fixed variable and removed it using a fixed comb filter designed for a specific FES frequency [15,16]. Although this method has provided new approaches for eliminating the M-wave from raw EMG signals during FES, it is difficult to obtain vEMG signals without the M-wave because their fixed cut-off frequencies cannot cope with the nonlinear properties of the M-wave. Therefore, various digital filters based on an adaptive algorithm or optimization method have been studied to consider the undefined dynamic characteristics of the M-wave, e.g., the hardware on-off control of amplifiers for eliminating stimulus artifacts [17], digital filtering algorithms using the comb filter [18], adaptive linear prediction filter [19], and notch filter combined with empirical mode decomposition [20] for eliminating the M-wave. However, a few of these approaches require additional devices, such as a blanking circuit that should be optimized to prevent the jittering issues caused by the complete separation of EMG and FES circuits. Moreover, it is difficult to formulate the M-wave as a mathematical model because of its nonlinear properties, which arbitrarily change with FES conditions, muscle fatigue, stimulation time, etc. In particular, it is technically difficult to use wearable FES systems because FES patterns should dynamically change to simultaneously reflect irregular and rapid changes in the behavior of patients because signal processing algorithms cannot completely eliminate the M-wave with dynamic variability [21]. Another difficulty is that the real-time capability of corresponding vEMG estimation algorithm. The comb filter and Gram–Schmidt filter are the only vEMG estimation algorithm applied to the real-time EMG-controlled FES system [22]. Most of the Gram–Schmidt filter methods were verified in static FES variables by using adaptive prediction error filter [23,24,25], and only the results of adjusting the stimulation frequency were presented by using adaptive match filter [22,26]. However, their validity has to be verified in dynamic FES conditions, because the variability of the M-wave can be significantly increased with a large change in the intensity of stimulation. In addition, these conditions must be considered in EMG-controlled FES systems for responding to rapid motion.

In this study, we propose a novel dual-channel EMG spatio-temporal differential (DESTD) method for effectively estimating vEMG signals during FES by eliminating stimulus artifacts and the M-wave under various stimulation environments of wearable FES systems. Figure 2a illustrates the components of the EMG signal measured when FES is applied to the muscle. The vEMG signal is a physiological signal that is independent of FES, and it can be modeled as a band-limited Gaussian random signal with a typical frequency bandwidth of 0–450Hz. The vEMG signal is physiologically generated by the motion intention, whereas the stimulus artifact and M-wave are generated only by FES. Therefore, the characteristics of the stimulus artifact and M-wave depend on the conditions of FES. The frequency characteristics of the stimulus artifact consist of the harmonics of the FES current. The M-wave is generated after the stimulus artifact and remains at approximately 10∼15 ms. On the basis of the above conceptual design of EMG signals during FES, we derived the DESTD method to estimate the vEMG signal by utilizing the physiological characteristics of FES-induced signals. Figure 2b illustrates the differences between physiological and FES-induced muscle contraction. Conventional FES-induced signal removal algorithms utilize only the signal characteristics of EMG signals, whereas the proposed method utilizes the physiological characteristics of the EMG signals of a stimulated muscle. We simultaneously measure two EMG signals for the same stimulated muscle and calculate the spatio-temporal difference between the signals. The proposed DESTD method is derived from a semi-simulation model that includes FES-induced and vEMG signals. An experimental study is conducted to verify the vEMG signal estimation performance, where the stimulation current is dynamically modulated by the proposed DESTD method.

## 2. Methods

### 2.1. Subjects

Five able-bodied healthy subjects (age: 50.0±5.2, weight: 69.3±12.8 kg, height: 166.8±6.6 cm) participated in this study. To ensure objective experimental results, the subjects were asked to avoid strenuous exercise 24 h before the experiment, and they performed the experiment while wearing comfortable clothes. This study was approved by the Public Institutional Review Board designated by the Ministry of Health and Welfare of Korea (P01-201909-13-002).

### 2.2. Experimental Setup

The experimental setup is shown in Figure 3. First, an EMG board comprising ADS-1299 (Texas Instruments [27]) and Nucleo-F767ZI (STMicroelectornics [28]) was designed. We used a commercially available EMG chip to sample a set of EMG data at 1 kHz and transferred the data to a computer through a USB interface. Second, an FES device (RehaStim, Hasomed) was utilized to apply electrical stimulation. Finally, a Kendall hydrogel electrode (Covidien) and ValuTrode electrode (Axelgaard Manufacturing) were utilized for EMG and FES, respectively.

### 2.3. Experimental Procedures

The gastrocnemius (GCM) muscle was selected for the experiments because it is easily contracted through an antigravity movement using the ankle joint. The experimental protocol was as follows: First, the subjects attached the EMG electrodes and performed an up-and-down movement while standing comfortably by slowly extending and flexing their ankles. This movement was performed approximately once per 5 s for 40 s. To ensure objective results, the data for the first and last 5 s were deleted, and the data for the remaining 30 s were utilized for analysis. Second, FES was applied to the GCM muscle to which the EMG electrodes were attached while sitting comfortably and relaxing the muscle. To verify the vEMG signal estimation performance of the DESTD method under dynamic FES conditions, the FES profile was designed as a trapezoidal pattern that increased for 0.5 s, was maintained for 1 s, decreased for 0.5 s, and repeated after 0.5 s. The EMG data measured via the above procedures were utilized to generate semi-simulated vEMG data with and without FES for verification. Finally, the second process was repeated after increasing the maximum stimulation intensity from 5 mA to 20 mA in increments of 5 mA. At more than 20 mA, the subjects began to feel uncomfortable pain. To ensure that the subjects were comfortable, the stimulation frequency and pulse width were set as 50 Hz and 250 μs, respectively. In addition, all data were filtered using a second-order band-pass filter of 20Hz to 500Hz and a notch filter of 60Hz harmonics to measure clean EMG data prior to applying the DESTD method.

Figure 4 shows the electrode attachment positions for comparative data analysis. In all experiments, each EMG signal was simultaneously measured using the conventional filters and DESTD method under the same stimulation conditions. The conventional filters for comparison with the proposed DESTD method were comb filter and Gram–Schmidt filter in consideration of real-time applicability. As shown in the Figure 4b, the inter-electrode distances for EMG for each method were set as 2 cm to meet the common standard for measuring bipolar surface EMG signals as much as possible [29]. In addition, the inter-electrode distance for FES was adjusted according to the GCM muscle size of each subject, and it was determined as 10 cm or more to generate sufficient muscle contraction.

### 2.4. Data Processing: DESTD Method

Volitional muscle contraction is physiologically generated from the brain because of the asynchronous contractions that occur in each muscle fiber. In contrast, FES-induced signals are manually generated by an FES device. These signals are synchronized with the frequency of FES because electrical impulses are simultaneously transmitted to every muscle fiber. The following assumptions can be made on the basis of these characteristics: The FES-induced EMG signals from all electrically stimulated muscles have the same tendency. In addition, the signals become more similar when the electrodes are more closely attached. This characteristic has been experimentally demonstrated in a previous study [30], although the experimental conditions are different from those used in this study. This assumption is quite useful because if two EMG electrodes are closely attached to a stimulated muscle, it is possible to simultaneously measure the same FES-induced signals and randomly different vEMG signals, which are generated because of the characteristic of Gaussian random signals, through each channel. Figure 5a shows the electrode positions in the proposed method. Note that two channels of EMG and one channel of FES are attached to the same muscle. The sensing and reference electrodes for EMG were shared to perform dual-channel EMG using the minimum number of electrodes. Moreover, a common ground electrode was attached between the stimulating and recording electrodes to reduce the amplitude of the stimulus artifact [31].

To derive a dual-channel vEMG signal estimation algorithm based on these electrode positions, we defined the raw EMG signal in the *j*th stimulation iteration as
(1)$$E_{j,1}\left(i\right)=V_{F,j,1}\left(i\right)+V_{V,j,1}\left(i\right)\;\mathrm{for}\;i=1,2,\ldots ,N$$
where $V_{F,j,1}$ and $V_{V,j,1}$ are the FES-induced EMG signal and voluntarily brain-induced vEMG signal, respectively, and *i* is the sampling index of EMG. Note that the continuous EMG signal can be divided into a constant number of samples because the electrical stimulation has a set of iterative patterns for a given stimulation frequency, as shown in Figure 5b. Therefore, *j* and *N* denote the stimulation iteration index of FES and the total number of samples between stimulation iterations, respectively. In addition, (Equation 1) is the raw EMG signal from EMG channel 1. Thus, the raw EMG signal from the other channel can be defined as
(2)$$E_{j,2}\left(i\right)=V_{F,j,2}\left(i\right)+V_{V,j,2}\left(i\right)\;\mathrm{for}\;i=1,2,\ldots ,N.$$

On the basis of the above equations, the main principle of the proposed dual-channel vEMG signal estimation algorithm is the removal of the FES-induced EMG signals by calculating the spatial difference between each EMG channel and the temporal difference between the following stimulation iterations. Note that the spatial difference is a simultaneous difference between EMG channels that are closely attached to the same stimulated muscle, and the temporal difference is a iterative difference between the successive stimulation. Thus, the above-mentioned spatial and temporal differences are defined as follows:(3)$$\left\{E_{j,1}\left(i\right)-E_{j,2}\left(i\right)\right\}-\left\{E_{j-1,1}\left(i\right)-E_{j-1,2}\left(i\right)\right\}=\bar{E_{j}}\left(i\right)-\bar{E_{j-1}}\left(i\right)=d\bar{E_{j}}\left(i\right)$$
where $\bar{E_{j}}\left(i\right)$ represents the spatial difference between the EMG channels in the *j*-th stimulation iteration and $d\bar{E}\left(i\right)$ represents the temporal difference in the derived spatial difference between the following stimulation iterations. Therefore, by substituting (Equation 3) and applying the same process to (Equation 1) and (Equation 2), it can be rewritten as
(4)$$d\bar{E_{j}}\left(i\right)=d\bar{V_{F,j}}\left(i\right)+d\bar{V_{V,j}}\left(i\right).$$

From (Equation 4), recall that the FES-induced EMG signals comprise the FES artifact and M-wave. Therefore, $d\bar{V_{F,j}}\left(i\right)$ can be defined as the summation of the FES artifact and M-wave, and (Equation 4) can be rewritten as
(5)$$d\bar{E_{j}}\left(i\right)=d\bar{V_{S,j}}\left(i\right)+d\bar{V_{M,j}}\left(i\right)+d\bar{V_{V,j}}\left(i\right)$$
where $V_{S,j}$ and $V_{M,j}$ denote the stimulus artifact and M-wave in the *j*-th stimulation iteration, respectively. The removal of FES-induced EMG signals and the estimation of the vEMG signal on the basis of (Equation 5) are described below.

#### 2.4.1. Stimulus Artifact EMG Signal

The stimulus artifact EMG signal was defined using the simulation model proposed in a previous study [32]. In general, the stimulus artifact is defined as a sequential signal such that a spike is maintained for a few milliseconds after electrical stimulation is delivered to a muscle, followed by a gradual decay. As the signal after electrical stimulation is the result of the rapid stimulation current measured through the resistance and capacitance properties of an EMG circuit, it is reasonable to define it as a gradually or exponentially decaying shape. In addition, the shape of the spike signal is determined by electrical stimulation pulses, such as monophasic or biphasic shapes. Generally, the magnitude of electrical stimulation is saturated by a clamping circuit because it typically exceeds the acceptable range of the voltage amplifier of the EMG circuit. When the spike is not clamped owing to a small electrical stimulation current, the magnitude is scaled according to the stimulation intensity.

On the basis of the above-defined simulation model, the stimulus artifact signal from EMG channel 1, $V_{S,j,1}$, can be defined as follows:(6)$$V_{S,j,1}\left(i\right)=K_{1}\;\cdot \;S_{j}\left(i\right)+\sum_{n=0}^{n_{k}}C_{j,1,n}\;\cdot \;e^{-\lambda_{j,n}i}$$
where $K_{1}$ is the magnitude scaling parameter of the spike and $S_{j}\left(i\right)$ is the shape function of the spike signal that is determined by the electrical stimulation pulse and maintained for a few milliseconds after the electrical stimulation. Note that $K_{1}$ is coerced between 0 and threshold value $K_{th}$ when the spike is saturated by the clamping circuit. Moreover, the exponential decay shape of the stimulus artifact that follows the spike signal is described as the sum of exponential functions using the stimulus artifact shape function fitting estimation method [33]. Note that $n_{k}$ is the number of exponential functions, and $C_{j,1,n}$ and $\lambda_{j,n}$ are the weight coefficients that determine the amplitude and decay rate parameter, respectively. The parameters and coefficients required to define the simulation model depend on the characteristics of the EMG circuit, and thus they should be experimentally determined during function-fitting procedures.

From (Equation 6), we assume that a measurement uncertainty exists between each EMG channel. As the principle of the EMG sensor is to detect the sum of the electric potentials measured in each muscle fiber, the measurement uncertainty between each channel is assumed as additive. In addition, we assume that the FES-induced EMG signals measured at each EMG channel have a subequal shape but different magnitudes because of the similarity of EMG signals measured from closely located electrodes on equally stimulated muscles. Therefore, $V_{S,j,2}\left(i\right)$ can be defined as
(7)$$V_{S,j,2}\left(i\right)=K_{2}\;\cdot \;S_{j}\left(i\right)+\sum_{n=0}^{n_{k}}C_{j,2,n}\;\cdot \;e^{-\lambda_{j,n}i}=\left(K_{1}+{△}_{k}\right)\;\cdot \;S_{j}\left(i\right)+\sum_{n=0}^{n_{k}}\left(C_{j,1,n}+{△}_{c}\right)\;\cdot \;e^{-\lambda_{j,n}i}$$
where $K_{2}=K_{1}+{△}_{k}$, $C_{j,2,n}=\left(C_{j,1,n}+{△}_{c}\right)$ and ${△}_{k}$ and ${△}_{c}$ are the additive measurement uncertainties of the spike and exponentially decaying signals, respectively, such that ${△}_{k}\neq 0$ and ${△}_{c}\neq 0$. Subtracting (Equation 6) from (Equation 7), we obtain
(8)$$\bar{V_{S,j}}\left(i\right)=V_{S,j,2}\left(i\right)-V_{S,j,1}\left(i\right)={△}_{k}\;\cdot \;S_{j}\left(i\right)+{△}_{c}\;\cdot \;\sum_{n=0}^{n_{k}}e^{-\lambda_{j,n}i}.$$

Note that the uncertainty between the EMG channels is caused by the distance between the two electrodes; therefore, it is equally applied to each stimulation iteration. Thus, subtracting (Equation 8) from the following stimulation iteration yields
(9)$$d\bar{V_{S,j}}\left(i\right)={△}_{k}\;\cdot \;\left\{S_{j}\left(i\right)-S_{j-1}\left(i\right)\right\}+{△}_{c}\;\cdot \;\left\{\sum_{n=0}^{n_{k}}\left(e^{-\lambda_{j,n}i}-e^{-\lambda_{j-1,n}i}\right)\right\}.$$

Moreover, the shape of the *j*-th stimulus artifact is predominantly modulated by muscle fatigue [34]. Thus, it is possible to assume that the difference in the shapes of the stimulus artifact between the following iterations is negligible, i.e., $S_{j}\left(i\right)=S_{j-1}\left(i\right)$ and $\lambda_{j,n}=\lambda_{j-1,n}$. On the basis of this assumption, the stimulus artifact can be removed by calculating the temporal difference between the following stimulation iterations of spatial differences between the EMG channels, regardless of the measurement uncertainties.

#### 2.4.2. M-Wave EMG Signal

The EMG signal of the M-wave is given using the simulation model of the M-wave [19], as follows:(10)$$V_{M,j,1}\left(i\right)=\alpha_{j,1}\;\cdot \;e^{-\frac{i}{\tau_{j,1}}}\;\cdot \;sin\left(\frac{2\pi 3i}{N}\right).$$
where $\alpha_{j,1}$ and $\tau_{j,1}$ are the scaling parameters that determine the amplitude and shape of the M-wave, respectively. We use the same assumptions as those applied in the previous section to define the shape-determining scaling parameter as a constant between the following stimulation iterations, i.e., $\tau_{j,1}=\tau_{j,2}=\tau$. There exists an additive measurement uncertainty for defining the amplitude of the M-wave, i.e., $\alpha_{j,2}=\alpha_{j,1}+{△}_{\alpha }$, where ${△}_{\alpha }$ is the additive measurement uncertainty, such that ${△}_{\alpha }\neq 0$. Therefore, subtracting $V_{M,j,1}$ from $V_{M,j,2}$ yields
(11)$$\begin{array}{c} \bar{V_{M,j}}\left(i\right)=V_{M,j,2}\left(i\right)-V_{M,j,1}\left(i\right)={△}_{\alpha }\;\cdot \;e^{-\frac{i}{\tau }}\;\cdot \;sin\left(\frac{2\pi 3i}{N}\right). \end{array}$$

As shown in (Equation 11), the spatial differences between the EMG channels can be assumed to be independent of the stimulation iterations. Therefore, the M-wave can be removed by calculating the spatio-temporal difference between the EMG channels by subtracting (Equation 11) from the following stimulation iteration.

#### 2.4.3. vEMG Signal

According to various previous studies [19], a voluntarily generated EMG signal can be defined as a band-limited Gaussian random signal. This is advantageous for the DESTD method, which linearly calculates spatial and temporal differences. First, in the process of calculating the spatial difference in the vEMG signal, i.e., $\bar{V_{V,j}}\left(i\right)=V_{V,j,2}-V_{V,j,1}$, $\bar{V_{V,j}}\left(i\right)$ can be considered as the vEMG signal estimated in the *j*-th stimulation iteration. This is because linearly subtracting two different band-limited Gaussian random signals yields a new band-limited Gaussian random signal, but it does not affect the frequency bandwidth and magnitude. In addition, the same approach can be applied to the temporal difference between the following stimulation iterations. Therefore, subtracting $\bar{V_{V,j}}\left(i\right)$ from the following stimulation iterations yields
(12)$$d\bar{V_{V,j}}\left(i\right)=\bar{V_{V,j}}\left(i\right)-\bar{V_{V,j-1}}\left(i\right)=V_{V,j}\left(i\right).$$

Consequently, the DESTD method can easily estimate the vEMG signal as follows:(13)$$\begin{array}{cc} \; & \left\{E_{j,1}\left(i\right)-E_{j,2}\left(i\right)\right\}-\left\{E_{j-1,1}\left(i\right)-E_{j-1,2}\left(i\right)\right\}=V_{V,j}\left(i\right). \end{array}$$

Note that (Equation 13) is derived from the assumption based on the physiological muscle recruitment mechanism that FES-induced muscle contractions are uniform over the stimulated muscle. Furthermore, as (Equation 13) is estimated using the various assumptions of the simulation models, it must be verified through practical experiments. Therefore, an experimental study was performed to validate the vEMG signal estimation performance of the DESTD method in practice.

### 2.5. Data Analysis

The raw EMG signal was generated during FES to verify the vEMG signal estimation performance of the DESTD method and compare it with that of the conventional methods. Recall that the components of the contaminated EMG signals during FES are the sum of the stimulus artifact, M-wave, and vEMG signal. Therefore, if each EMG signal is measured separately and combined, the contaminated raw EMG signal during FES can be semi-simulated. As shown in Figure 6, the raw EMG signal during FES (Figure 6c) was semi-simulated by adding the EMG data with volitional effort measured without FES (Figure 6a) to the EMG data without volitional effort measured during FES (Figure 6b). This method provides the advantage of obtaining objective comparators from already contaminated EMG signals, and it has been utilized in previous studies [22,26]. Moreover, note that the EMG data were normalized between 0 and 1 on the basis of the case when the stimulation current was 20mA.

The semi-simulated raw EMG signal during FES was utilized to conduct data analysis. The reference vEMG signal for comparison was defined as the vEMG signal measured without FES, which is shown in Figure 6a. This is reasonable because if the algorithm is correctly applied, the stimulus artifact and M-wave should be completely removed, and the filtered vEMG signal should match the EMG signal without FES. For quantitative comparison, the root mean square (RMS) values of the vEMG signal estimated under each FES condition were compared with those of the reference vEMG signal by calculating the normalized root mean square error (NRMSE). The RMS values are defined as follows:(14)$$RMS_{C}\left(i\right)=\sqrt{\frac{1}{m}\sum_{j=i-m}^{i}V_{V,C}{\left(j\right)}^{2}}\;\mathrm{for}\;i=m+1,\;cdots,N$$
and
(15)$$RMS_{D}\left(i\right)=\sqrt{\frac{1}{m}\sum_{j=i-m}^{i}V_{V,D}{\left(j\right)}^{2}}\;\mathrm{for}\;i=m+1,\cdots ,N$$
where $RMS_{C}\left(i\right)$ and $RMS_{D}\left(i\right)$ represent the calculated RMS values for the conventional filters and DESTD method, respectively; *i* denotes the sampled time index; *m* denotes the window length, which is defined as 1000 samples in this experiment; *N* denotes the total number of samples; and $V_{V,\;\cdot \;}\left(j\right)$ represents the vEMG values estimated by each algorithm. In addition, the NRMSE is defined as follows:(16)$$NRMSE_{C}=\frac{\sqrt{\frac{1}{N}\sum_{j=1}^{N}{\left\{RMS_{C}\left(j\right)-RMS_{R}\left(j\right)\right\}}^{2}}}{\left|RMS_{C,max}-RMS_{C,min}\right|}$$
and
(17)$$NRMSE_{D}=\frac{\sqrt{\frac{1}{N}\sum_{j=1}^{N}{\left\{RMS_{D}\left(j\right)-RMS_{R}\left(j\right)\right\}}^{2}}}{\left|RMS_{D,max}-RMS_{D,min}\right|}$$
where $RMS_{R}\left(j\right)$ represents the RMS value of the reference vEMG signal measured without FES, and $RMS_{\;\cdot \;,max}$ and $RMS_{\;\cdot \;,min}$ are the maximum and minimum values of each RMS value, respectively.

### 2.6. Statistical Analysis

For the quantitative verification of the DESTD method, a two-tailed two-sample t-test was conducted to compare the vEMG signal estimation performances of the comb filter, Gram–Schmidt filter, and DESTD method. In addition, two-way analysis of variance was performed for the filtering method and stimulation intensity. Every 5 subjects performed the same 3 tests for each 5 stimulation conditions. As mentioned in the experimental procedure section, only the experimental data for 30 s were used, and an average of six voluntary muscle contractions occurred per experiment. Therefore, the number of trials per person under one experimental condition was approximately 90. The significance level was set as 0.01.

## 3. Results and Discussion

### 3.1. Results

Figure 7 shows an example of the EMG experimental results obtained when the rapidly changing FES with a maximum intensity of 20 mA was applied to the GCM muscle. As shown in Figure 7a, the vEMG signal was completely covered by the FES artifact and M-wave. Figure 7c,d show the vEMG data estimated from the contaminated EMG data by the comb filter, Gram–Schmidt filter, and DESTD method, respectively, with FES. The DESTD method effectively removed the FES-induced signals regardless of the FES pattern compared to the comb filter and Gram–Schmidt filter. Figure 7b shows the FES-induced signal removal performances for each filter in more detail. When the stimulus intensity remained constant, all filters effectively removed the FES-induced signals. However, when the stimulus intensity changed, the conventional filters could not completely remove the FES-induced signals. In contrast, the DESTD method successfully removed the FES-induced signals while retaining the vEMG data.

Table 1 shows the quantitative comparisons of the vEMG signal estimation performance with respect to the maximum stimulation current. As shown in Table 1, the NRMSE between the vEMG estimated through each filter and reference vEMG measure in the absence of FES were calculated according to the maximum stimulation currents of FES, and the t-test and ANOVA analysis were conducted. First, as a result of ANOVA analysis to analyze the effect of filter type or maximum stimulation current, the results revealed that both factors have significant effects on NRMSE ($F=81.12,\;p=7.8598\times 10^{-28}$ for the filter type and $F=41.85,\;p=8.4864\times 10^{-27}$ for the maximum stimulation current). In addition, as a result of analyzing the effect of the only maximum stimulation current on NRMSE according to the filter type, it revealed that there was significant effect on the comb filter ($F=57.25,\;p=1.0271\times 10^{-22}$) and the Gram–Schmidt filter ($F=36.11,\;p=3.2092\times 10^{-17}$), but no effect on the DESTD method (F=0.31, p=0.871). These results confirm that the proposed DESTD method can estimate vEMG robustly regardless of the maximum current of dynamically modulating stimulation compared to the comb filter and Gram–Schmidt filter. Moreover, the t-test was conducted to validate the vEMG estimation performances of each filter by comparing the vEMGs estimated under non-stimulating condition with those estimated under various stimulating conditions. Note that as shown in the results under 0 mA condition of Table 1, the estimated vEMGs from each filter are similar enough to the reference clean EMG with sufficiently low NRMSE values. This is also shown in Figure 8b in which the RMS values of the reference vEMG, vEMG data through comb filter, Gram–Schmidt filter, and DESTD method are similar. When the FES is applied, however, the NRMSE values of vEMG data through comb filter and Gram–Schmidt filter were increased with significant differences ($p=0.0250,\;4.3541\times 10^{-7},\;2.0478\times 10^{-12}$ and $1.3654\times 10^{-13}$ by comb filter, and $p=0.4785,\;9.6536\times 10^{-4},2.8649\times 10^{-8}$ and $1.2714\times 10^{-10}$ by Gram–Schmidt filter for maximum 5 ,10 ,15, and 20 mA stimulation current, respectively) as shown in Figure 8c. Note that the Gram–Schmidt filter showed good performance under only 5 mA stimulation condition without statistical difference from the data under non-stimulating condition. In contrast to these results, the proposed DESTD method successfully estimated vEMG with remaining small NRMSE regardless of stimulation currents (p=0.7911, 0.6075, 0.6282 and 0.2581 for maximum 5, 10, 15, and 20 mA stimulation current, respectively). The overall NRMSE comparison results are presented in Figure 8a. From the above results, it can be inferred that the DESTD method is more robust against modulation of stimulation conditions compared to the conventional comb filter and Gram–Schmidt filter.

Figure 9 shows the frequency analysis for the same example. The magnitude of the harmonics of the raw EMG data was extremely large at the stimulation frequency (50 Hz). This magnitude must be reduced to eliminate the FES-induced EMG signals. Although the comb filter and Gram–Schmidt filter were designed to remove FES artifacts, it could not completely remove the harmonics at 50 Hz. In contrast, the DESTD method effectively removed the harmonics at the stimulation frequency. Thus, the DESTD method successfully eliminated the FES-induced EMG signals.

### 3.2. Discussion

The main goal of estimating vEMG signals is to remove the stimulus artifact and M-wave. Despite extensive research, FES-induced signals have not been completely eliminated owing to the difficulty in the mathematical modeling of each component and the nonlinearity of the M-wave, which is particularly deteriorated in dynamic stimulation conditions. This motivated us to develop a method for robustly detecting vEMG signals even under dynamic FES conditions.

The main advantage of the DESTD method is that the calculation processes are simple and intuitive. This simplifies the systems to which the method is applied. The system simplicity is particularly advantageous for portable or wearable systems. Note that the EMG sensor (Figure 3) used in the experiment was designed to be larger than necessary because it is a general purpose device for receiving 8-channel EMG signals in addition to the purpose of this study. We used an integrated EMG and FES board with a weight of 14 g and dimensions of 45*35*14[mm]. A wearable EMG-controlled FES system with this integrated board was designed to perform a small-scale clinical trial study by commissioned research [35]. 29 elderly subjects (age: 75.0±5.36, weight: 62.35±8.6 kg, height: 156.16±8.6 cm) participated under two randomly assigned conditions (with and without EMG-controlled FES). The performance of the subjects was improved in terms of gait speed (11.1%), cadence (15.6%, p<0.01), GCM symmetry ratio (9.9% in the stance phase and 11.8%, p<0.05 in the swing phase), and balance (10.7%, p<0.01).

The DESTD method has certain limitations. First, the range of the frequency of FES is limited by the sampling frequency of the EMG sensor board and the duration for which the stimulus artifact remains, which is typically approximately 1∼4 ms. The DESTD method involves the division of continuous EMG data into a certain number of samples and the calculation of the difference between adjacent stimulation iterations. In this process, the FES and EMG sampling frequencies determine the division of continuous EMG data. The performance of the DESTD method can degrade if the stimulation frequency is higher than the EMG sampling frequency. In this study, an FES frequency of under 50 Hz and an EMG sampling frequency of over 1 kHz were required to achieve an acceptable vEMG signal estimation performance. Second, the area where the electrodes are attached to the muscle should be wide because of the use of dual-channel electrodes for EMG and single-channel electrodes for FES. Note that we used a custom manufactured electrode set with dimensions of 5*16[cm]. Thus, the proposed system can be only utilized for large muscles such as the lower extremity muscles, i.e., the GCM, tibialis anterior, rectus femoris, and biceps femoris muscles. Therefore, a method for minimizing the size of electrodes should be developed by optimizing electrode usage. Last, there should be more subjects to support the experimental results. In this paper, a large number of experiments were conducted per subject to overcome this limitation, but more subjects must be recruited in order to obtain objective results with sufficient statistical importance.

## 4. Conclusions

A novel vEMG signal estimation method is proposed to estimate the volitional muscle activity of the muscle to which FES is applied. The proposed DESTD method utilizes two EMG sensors and calculates the spatial and temporal difference between each EMG sensor to remove FES-induced EMG signals. The DESTD method can estimate vEMG signals with or without FES. Furthermore, it can ensure good estimation performance in static or dynamic FES conditions. The vEMG signal estimation performance of conventional algorithms, such as comb filters, adaptive filters, decomposition methods, and Gram–Schmidt filter, is either unverified or limited under dynamic FES conditions. Therefore, the DESTD method can be used in various applications where EMG and FES are simultaneously and dynamically utilized, such as wearable EMG-controlled FES systems.

Unlike conventional algorithms, the DESTD method estimates vEMG signals by utilizing the physiological characteristics of EMG signals from a spatio-temporal viewpoint. The method utilizes the physiological difference between FES-induced and volitional EMG signals, and it effectively removes unnecessary signals from the raw EMG signals contaminated by FES. The DESTD method is verified by modeling a semi-simulated raw EMG signal when FES is applied. The effectiveness of the DESTD method is experimentally verified. The experimental results of the vEMG signal estimation performance of the DESTD method and conventional comb filter and Gram–Schmidt filter are compared. The DESTD method can estimate vEMG signals more effectively than the conventional filters. Regardless of rapid changes in the stimulation intensity, the DESTD method removes FES-induced EMG signals to a large extent while retaining brain-induced vEMG signals.

In future work, the DESTD method will be applied to other muscles. For this purpose, the proposed electrode attachment positions should be optimized such that the electrodes occupy a small space. A method in which an electrode is shared between FES and EMG is being developed for the proposed application. Moreover, a wearable EMG-controlled FES system with the DESTD method will be studied and applied to various circumstances. Such a system has good potential for application in the clinical and rehabilitation fields because the motion intention of patients can be used to directly control the amplitude of the stimulation intensity. In particular, the system has the potential to improve the muscle contraction efficiency in our present work [35]. For these reasons, clinical trials will be conducted to investigate the clinical and rehabilitation advantages of the proposed wearable EMG-controlled FES system on a large scale.

## Figures and Tables

**Figure 1 sensors-21-08015-f001:**
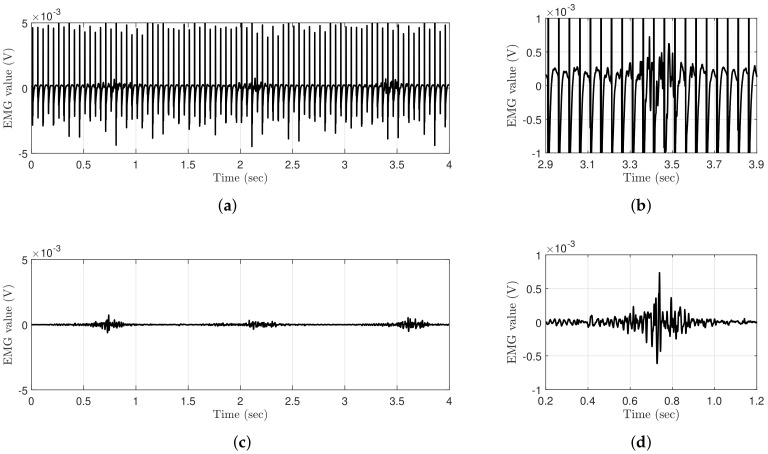
Comparison of raw EMG signals where the muscle is voluntarily and periodically contracted with and without FES. (**a**) Raw EMG signal with FES. (**b**) Raw EMG signal with FES (magnified). (**c**) Raw EMG signal without FES. (**d**) Raw EMG signal without FES (magnified).

**Figure 2 sensors-21-08015-f002:**
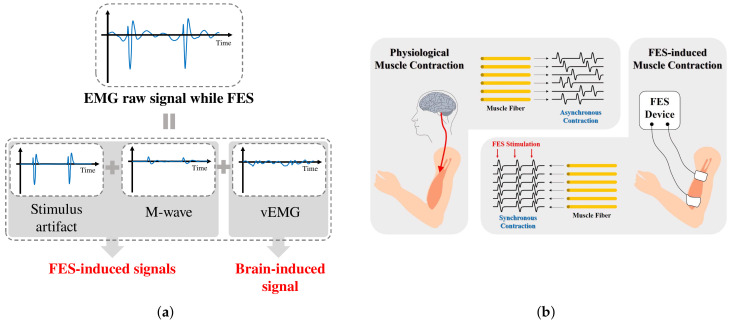
Conceptual design of EMG signals during FES: (**a**) Components of contaminated EMG signal during FES and (**b**) comparison of physiological muscle contraction and FES-induced muscle contraction.

**Figure 3 sensors-21-08015-f003:**
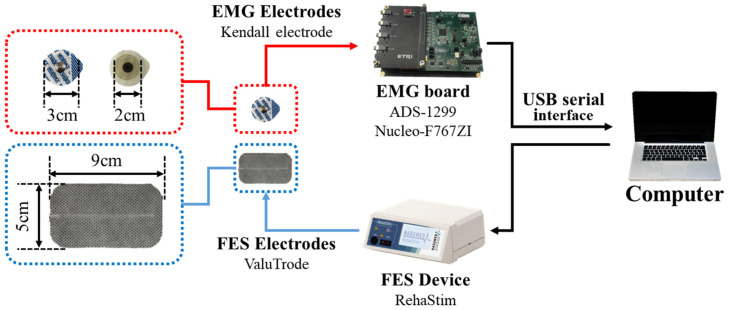
Schematic of experimental setup.

**Figure 4 sensors-21-08015-f004:**
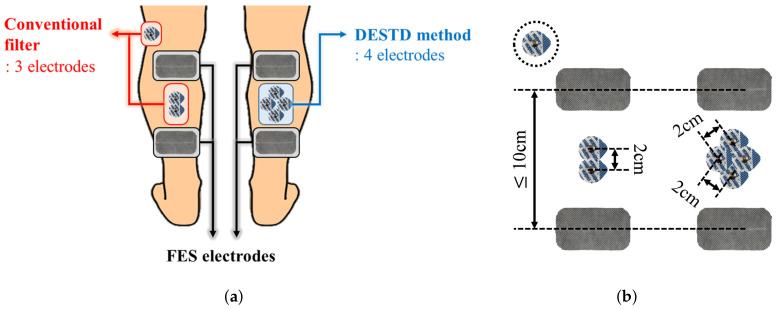
Electrode attachment positions for comparative data analysis: (**a**) Comparison of electrode attachment positions between conventional filter and DESTD method and (**b**) inter-electrode distances for each method.

**Figure 5 sensors-21-08015-f005:**
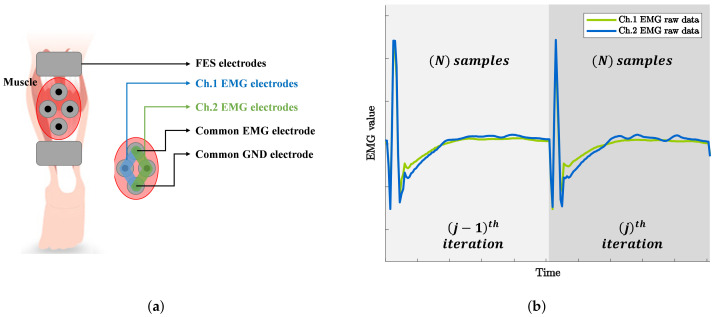
Dual-channel EMG data acquisition system: (**a**) Electrode attachment positions and (**b**) example of raw EMG data recorded from each EMG channel under electrical stimulation.

**Figure 6 sensors-21-08015-f006:**
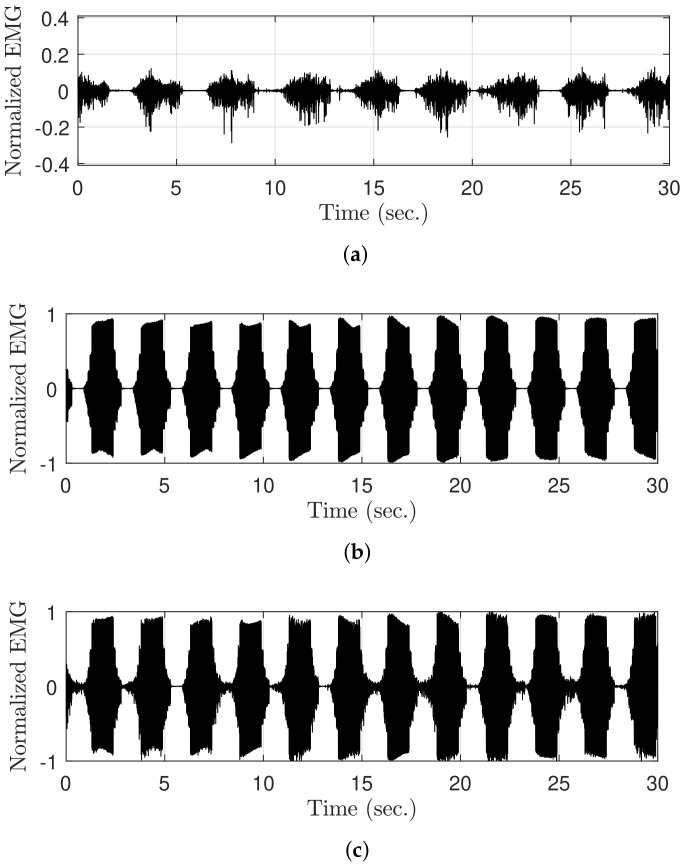
Semi-simulated raw EMG data during FES for the verification of the DESTD method: (**a**) Raw EMG data with volitional effort, (**b**) raw EMG data without volitional effort during FES, and (**c**) semi-simulated raw EMG data with volitional effort during FES.

**Figure 7 sensors-21-08015-f007:**
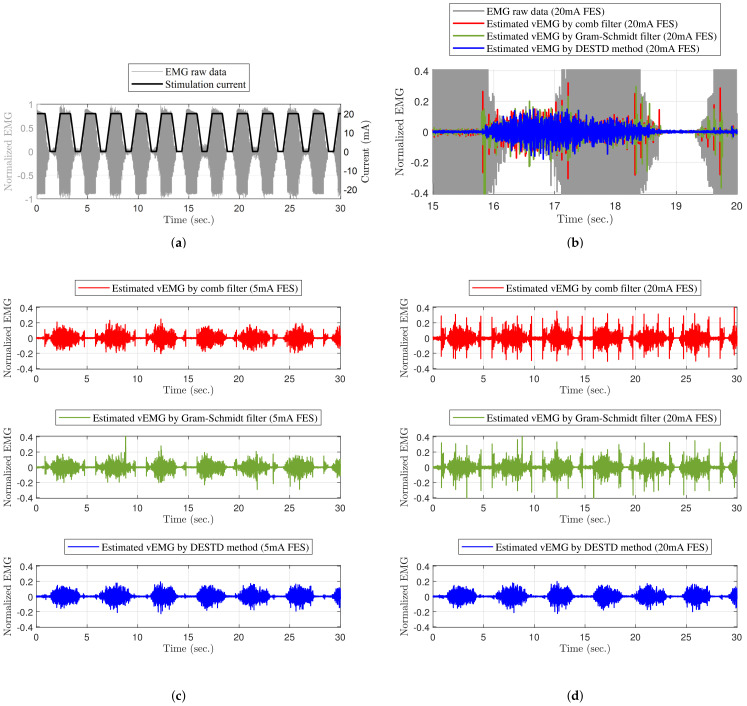
Example of EMG experimental results: (**a**) stimulation current and EMG raw data without filters, (**b**) vEMG comparison of experimental results over 15 to 20 s, vEMG data filtered by filters with maximum (**c**) 5 mA, and (**d**) 20 mA stimulation current of dynamically modulating stimulation.

**Figure 8 sensors-21-08015-f008:**
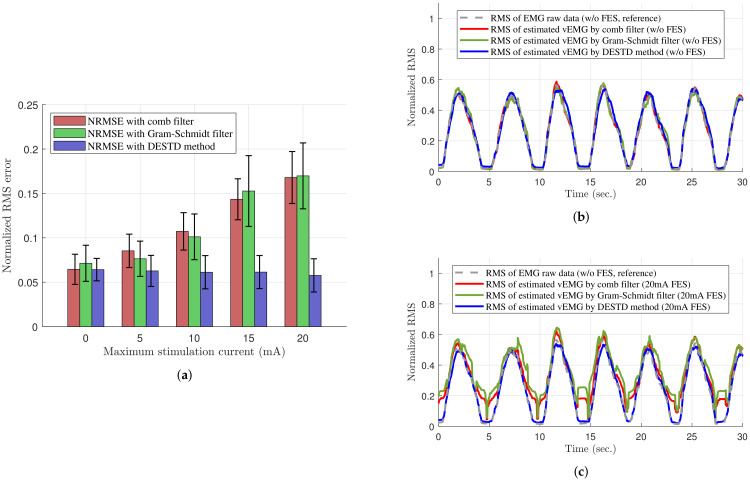
Performance of removing the FES-induced signals of the comb filter, Gram–Schmidt filter, and DESTD method. (**a**) NRMSE values of the estimated vEMGs; (**b**) RMS values of the estimated vEMGs without FES; and (**c**) those with FES of 20 mA maximum stimulation current using the comb filter, Gram–Schmidt filter, and DESTD method.

**Figure 9 sensors-21-08015-f009:**
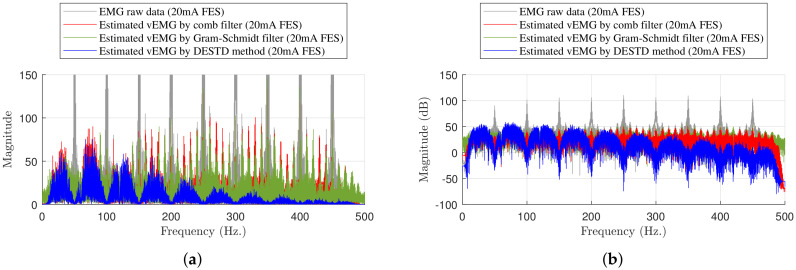
Frequency analysis example for experimental results at maximum stimulation intensity of 20 mA: (**a**) Comparison of experimental results and (**b**) the result in dB Scale.

**Table 1 sensors-21-08015-t001:** Experimental results: NRMSE of the filtered EMG data.

Stimulation	Filter	Subjects					Overall
Current (mA)		Subject 1	Subject 2	Subject 3	Subject 4	Subject 5	Mean ± std
0mA	Comb filter	0.0556 ± 0.0083	0.0826 ± 0.0129	0.0577 ± 0.0137	0.0694 ± 0.0157	0.0551 ± 0.01332	0.0645 ± 0.0170
	Gram-Schmidt filter	0.0655 ± 0.0056	0.0919 ± 0.0215	0.0565 ± 0.0132	0.0813 ± 0.0161	0.0563 ± 0.0102	0.0713 ± 0.0203
	DESTD method	0.0557 ± 0.0101	0.0747 ± 0.0138	0.0642 ± 0.0086	0.0711 ± 0.0075	0.0567 ± 0.0055	0.0641 ± 0.0126
5mA	Comb filter	0.0730 ± 0.0046	0.0960 ± 0.0154	0.0771 ± 0.0092	0.1057 ± 0.0204	0.0707 ± 0.0134	0.0853 ± 0.0187 ^**^
	Gram-Schmidt filter	0.0619 ± 0.0149	0.0985 ± 0.0252	0.0679 ± 0.0025	0.0690 ± 0.0030	0.0784 ± 0.0045	0.0763 ± 0.0198
	DESTD method	0.0528 ± 0.0110	0.0763 ± 0.0147	0.0503 ± 0.0194	0.0811 ± 0.0016	0.0513 ± 0.0054	0.0627 ± 0.0174
10mA	Comb filter	0.0931 ± 0.0037	0.1163 ± 0.0192	0.1017 ± 0.0051	0.1277 ± 0.0289	0.0996 ± 0.0137	0.1072 ± 0.0210 ^**^
	Gram-Schmidt filter	0.0841 ± 0.0124	0.1021 ± 0.0400	0.1099 ± 0.0047	0.0947 ± 0.0159	0.1173 ± 0.0153	0.1011 ± 0.0257 ^**^
	DESTD method	0.0437 ± 0.0164	0.0785 ± 0.0177	0.0587 ± 0.0068	0.0750 ± 0.0054	0.0523 ± 0.0054	0.0612 ± 0.0187
15mA	Comb filter	0.1249 ± 0.0094	0.1482 ± 0.0260	0.1348 ± 0.0023	0.1741 ± 0.0153	0.1382 ± 0.0135	0.1434 ± 0.0230 ^**^
	Gram-Schmidt filter	0.1394 ± 0.0157	0.1506 ± 0.0696	0.1556 ± 0.0077	0.1584 ± 0.0341	0.1622 ± 0.0126	0.1526 ± 0.0398 ^**^
	DESTD method	0.0434 ± 0.0157	0.0794 ± 0.0139	0.0581 ± 0.0109	0.0770 ± 0.0036	0.0512 ± 0.0045	0.0614 ± 0.0186
20mA	Comb filter	0.1503 ± 0.0083	0.1739 ± 0.0404	0.1585 ± 0.0033	0.1887 ± 0.0355	0.1684 ± 0.0164	0.1679 ± 0.0292 ^**^
	Gram-Schmidt filter	0.1518 ± 0.0184	0.1590 ± 0.0589	0.1764 ± 0.0215	0.1695 ± 0.0248	0.1952 ± 0.0154	0.1697 ± 0.0371 ^**^
	DESTD method	0.0374 ± 0.0097	0.0777 ± 0.0087	0.0481 ± 0.0179	0.0746 ± 0.0030	0.0498 ± 0.0042	0.0576 ± 0.0187

The significant different between the NRMSE of EMG data filtered by each algorithm is indicated by 1 asterisk (p<0.05) or 2 asterisks (p<0.01). The table represents the individual and overall results of each vEMG estimation performance as the stimulation condition varies, respectively.

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
