# Peer review of "Volitional EMG Estimation Method during Functional Electrical Stimulation by Dual-Channel Surface EMGs"

_sensors, 2021, doi:10.3390/s21238015_

Round 1

Reviewer 1 Report

The paper presents. novel method for EMG signal processing aiming at an accurate stimulation of the vEMG. The overall paper is well presented and results are quantitatively evaluated. The conclusion is in adequation with the results.

However, this submission needs minor changes before publication:

  • please, perform a careful proof reading for remaining typos (for instance at line 166 'rangdom')
  • Figure 3 clarify the authors objectives, and also clarifies the Figure 1, but comes late in the explanation. This clear vision should be more visible in the introduction.
  • Introduction should be improved, especially the state of the art on vEMG estimation is low  (citations 12 to 19), as it is the principal objective of the paper. 
  • This state of the art should be used in the result section where the authors compare their method with comb filters, but results are not discussed or compared to methods reported in the introductive state of the art.
  • In Figure 7a, include both curves in legend
  • In figure 9, y-axis should be in log scale to provide a overall comparison of provided curves without  compromising figure size.

Author Response

 Thank you for such invaluable comments. In the revised paper, significant changes are in yellow highlights. Please kindly understand that it was difficult to mark every minor change in English after receiving an overall revision of this article from an expert. According to the reviewers’ suggestions, we have revised the paper.

 Please refer to the attached file for detailed answers.

Reviewer 2 Report

This paper presents a new two-channel EMG time-space differential method for vEMG signal estimation during FES. It is novel and the results are valuable for reference. The detailed comments are laid out below:

1.Lines 36-46 introduced "two types of wearable FES systems integrating with EMG sensors". It is recommended to add references to support the introduction part in this section.

2.Line 41 mentioned that "due to difficulty", the specific difficulty should be specifically pointed out.

3.In Lines 61-62, "some researchers assumed M-waves as a fixed variable". It is recommended to supplement relevant reference.

4.Lines 119-121, it is recommended to add the description and schematic diagram of the motor placement in section 2.3 instead of in section 2.4 to increase clarity.

5.In Line 123, 30s experimental data of 5 experimenters were studied, but the samples were not huge enough. Although it was mentioned that "sufficient statistical importance was shown", more samples should be included if possible.

6.Regarding equation 3, the expression of "the removal of FES-induced EMG signals by calculating the spatial difference between each EMG channel and the temporal difference between the following stimulus iterations", as well as the meaning of the spatial difference and the time difference should also be explained more specifically to increase clarity.

7.K2 in Equation 7 was not defined.

8.In Lines 257-261, 3 experiments were conducted on 5 experimenters, and the number of repetitions was set to 15. The expression of "repetitions" is not clear here, please explain clearly.

9.Experimental results under 20ma stimulation was shown in the paper. It is recommended to supplement the results under 5ma stimulation to reflect the superiority of 20ma stimulation and to support "the comb filter mentioned cannot completely remove the FES-induced signal when the stimulus changes" in Line 274.

Author Response

(The authors gave the same response as above.)

Round 2

Reviewer 2 Report

After correcting the grammatical/sentence errors in the article, it is recommended to publish.